# What Matters in Learning from Offline Human Demonstrations for Robot Manipulation

**Ajay Mandlekar**[1], **Danfei Xu**[1], **Josiah Wong**[1], **Soroush Nasiriany**[2], **Chen Wang**[1],

**Rohun Kulkarni**[1], **Li Fei-Fei**[1], **Silvio Savarese**[1], **Yuke Zhu**[2], **Roberto Martín-Martín**[1]

[1]Stanford University, [2]The University of Texas at Austin

**Abstract:** Imitating human demonstrations is a promising approach to endow robots with various manipulation capabilities. While recent advances have been made in imitation learning and batch (offline) reinforcement learning, a lack of open-source human datasets and reproducible learning methods make assessing the state of the field difficult. In this paper, we conduct an extensive study of six offline learning algorithms for robot manipulation on five simulated and three real-world multi-stage manipulation tasks of varying complexity, and with datasets of varying quality. Our study analyzes the most critical challenges when learning from offline human data for manipulation. Based on the study, we derive a series of lessons including the sensitivity to different algorithmic design choices, the dependence on the quality of the demonstrations, and the variability based on the stopping criteria due to the different objectives in training and evaluation. We also highlight opportunities for learning from human datasets, such as the ability to learn proficient policies on challenging, multi-stage tasks beyond the scope of current reinforcement learning methods, and the ability to easily scale to natural, real-world manipulation scenarios where only raw sensory signals are available. We have open-sourced our datasets and all algorithm implementations to facilitate future research and fair comparisons in learning from human demonstration data at https://arise-initiative.github.io/robomimic-web/

**Keywords:** Imitation Learning, Offline Reinforcement Learning, Manipulation

## 1 Introduction

Human supervision has been at the heart of the most significant recent advances in several domains such as computer vision [1–4] and natural language processing [5–7]. By intelligently extracting information from large-scale human-labeled datasets, autonomous machines have been able to reach near- or even super-human performance on decades-old problems such as image recognition and question answering. Roboticists have also attempted to tackle robot manipulation through learning from human datasets, using the paradigms of Imitation Learning [8–10] and Batch (Offline) Reinforcement Learning [11–13], where datasets consisting of robot arm trajectories, action labels at each timestep, and possibly reward labels, are used to train closed-loop policies.

As in other domains, large offline datasets offer several benefits such as scale, portability, and reproducible evaluations to measure progress. Recently, there has been considerable progress in offline learning for robot manipulation from human demonstrations [9, 14, 10]. Despite these advances, the offline learning paradigm has not been nearly as disruptive in robotics as in other disciplines – there is a large gap between autonomous robot manipulation capabilities and the wide range of tasks that humans can solve effortlessly using physical and cognitive intelligence. What has inhibited the use of large human-provided datasets to address this gap?

In contrast to other domains where supervised learning has been successful, robotic manipulation is a time-evolving dynamical system, requiring fine-grained real-time control to guide robot arms successfully through tasks – consequently, data collection can present technical challenges requiring

5th Conference on Robot Learning (CoRL 2021), London, UK.

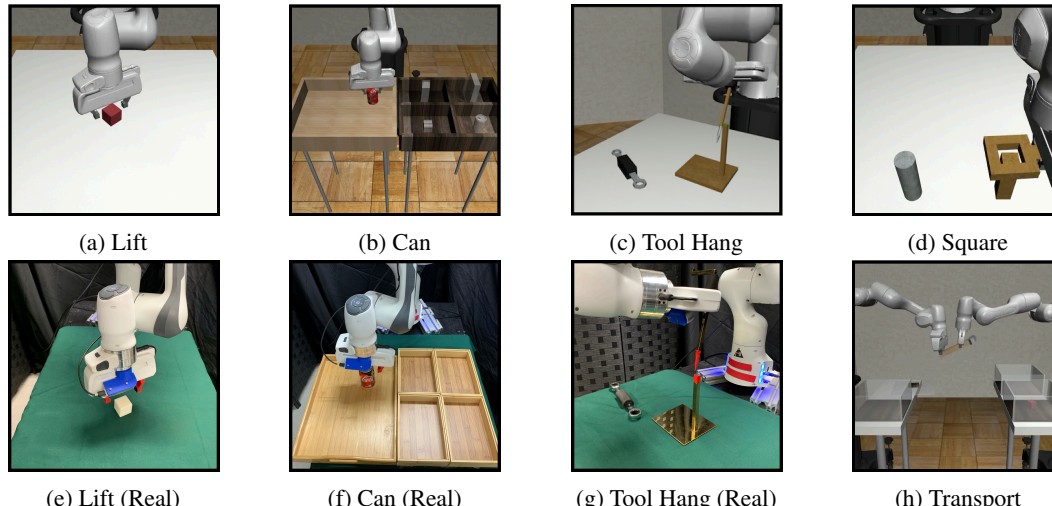

| (a) Lift | (b) Can | (c) Tool Hang | (d) Square |
| (e) Lift (Real) | (f) Can (Real) | (g) Tool Hang (Real) | (h) Transport |

Figure 1: **Tasks.** We collect datasets across 6 operators of varying proficiency and evaluate offline policy learning methods on 8 challenging manipulation tasks that test a wide range of manipulation capabilities including pick-and-place, multi-arm coordination, and high-precision insertion and assembly.

specialized systems [15], which can explain why large-scale human-provided datasets [16, 17] have not been very prevalent. Learning from such datasets can also present several challenges. Human demonstrations can differ from machine-generated datasets (a recent trend in benchmarks for offline policy learning [18, 19]) due to a non-Markovian decision process, since humans may not act purely based on the current observation. There can also be significant variance in both data quality and solution strategy when collecting data from multiple humans [20]. Differences from classic supervised learning, such as a mismatch between training and evaluation objectives (task success rate), can make selecting a final policy challenging [21, 22], especially in real-world settings where evaluating each policy on a robot can be infeasible. Finally, offline learning is sensitive to state and action space coverage (dataset size) and agent design decisions.

Studying these challenges in the context of robot manipulation and human-provided datasets could be a stepping stone to closing the gap between robot and human manipulation capabilities. Unfortunately, a lack of suitable benchmark and human datasets have made studying this setting difficult. Prior works are either limited to studying simple 2D environments [23] or using data generated from hard-coded policies [24, 25]. In this paper, we address this need by presenting a study of data-driven offline policy learning methods on several human-provided robot manipulation datasets. We collect task demonstrations from human teleoperators across a broad range of simulated and real world manipulation tasks and investigate several factors that play a role in learning from such data.

From our results, we point out several lessons to guide future research in leveraging human supervision for robot manipulation effectively. We find that history-dependent models can be extremely effective in learning from single and multi-human datasets while state-of-the-art batch RL algorithms struggle to learn from such datasets, and that the choice of observation space and hyperparameters play a substantial role in training proficient policies. We also find that there is substantial promise for solving more complex tasks using large-scale human datasets and that our insights directly transfer to real-world scenarios, making this an important setting to explore further.

## 2   Challenges in Offline Learning from Human Datasets

In this section, we outline five challenges in offline learning from human datasets that motivate different factors that we investigate in our study.

**(C1) Data from Non-Markovian Decision Process.** Human demonstrations can differ substantially from machine-generated demonstrations because humans may not act purely based on a single current observation. External factors (teleoperation device, past actions, history of episode) may all play a role. Prior work [20] has noted substantial benefits from leveraging models that are history-dependent and / or with temporal abstraction to learn from human demonstrations. We investigate various design choices related to such architectures in this study.

| Dataset | BC | BC-RNN | BCQ | CQL | HBC | IRIS |
|---|---|---|---|---|---|---|
| Lift (MG) | 65.3±2.5 | 70.7±3.4 | **91.3±1.9** | 64.0±2.8 | 47.3±4.1 | **96.0±1.6** |
| Can (MG) | 64.7±3.4 | 68.7±2.5 | **75.3±0.9** | 1.3±0.9 | 40.7±3.4 | 48.0±6.5 |
| Lift (PH) | **100.0±0.0** | **100.0±0.0** | **100.0±0.0** | 92.7±5.0 | **100.0±0.0** | **100.0±0.0** |
| Can (PH) | 95.3±0.9 | **100.0±0.0** | 88.7±0.9 | 38.0±7.5 | **100.0±0.0** | **100.0±0.0** |
| Square (PH) | 78.7±1.9 | **84.0±0.0** | 50.0±4.9 | 5.3±2.5 | 82.6±0.9 | 78.7±2.5 |
| Transport (PH) | 17.3±2.5 | **71.3±6.6** | 7.3±3.3 | 0.0±0.0 | 48.6±3.8 | 41.3±3.4 |
| Tool Hang (PH) | **29.3±0.9** | 19.3±5.0 | 0.0±0.0 | 0.0±0.0 | 30.0±7.1 | 11.3±2.5 |
| Lift (MH) | **100.0±0.0** | **100.0±0.0** | **100.0±0.0** | 56.7±40.3 | **100.0±0.0** | **100.0±0.0** |
| Can (MH) | 86.0±4.3 | **100.0±0.0** | 62.7±8.2 | 22.0±5.7 | 91.3±2.5 | 92.7±0.9 |
| Square (MH) | 52.7±6.6 | **78.0±4.3** | 14.0±4.3 | 0.7±0.9 | 60.7±5.0 | 52.7±5.0 |
| Transport (MH) | 11.3±2.5 | **65.3±7.4** | 2.6±0.9 | 0.0±0.0 | 14.0±1.6 | 10.7±0.9 |

Table 1: **Results on Low-Dimensional Observations.** We present success rates averaged over 3 seeds for each method across the low-dim Machine-Generated (MG), Proficient-Human (PH), and Multi-Human (MH) datasets. The results show that methods that model temporal correlations (BC-RNN, HBC, IRIS) exhibit strong performance on human datasets. Furthermore, while Batch RL algorithms like BCQ are proficient on machine-generated data, they perform poorly on human datasets.

**(C2) Variance in Demonstration Quality from Multiple Humans.** Prior work [20, 17] has found that data collected from several humans can differ substantially in both demonstration proficiency and solution strategy. Differences in supervisor proficiency can manifest in many ways, such as large variations in trajectory length and noise in robot movement or mistakes (e.g. missed grasps). In our study, we evaluate offline policy learning algorithms on such datasets. While recent batch RL algorithms have shown an excellent ability to learn from mixed quality machine-generated datasets [26, 27], we empirically find that they fail to learn well from mixed quality human data.

**(C3) Dependence on Dataset Size.** Offline policy learning is sensitive to the state and action space coverage in the dataset, and by extension, the size of the dataset itself. In our study, we investigate how dataset sizes affect policy performance. This analysis is useful to understand the value of adding more data – an important consideration since collecting human demonstrations can be costly.

**(C4) Mismatch between Training and Evaluation Objectives.** Unlike traditional supervised learning, where model selection can be achieved by using the model with the lowest validation loss [21], offline policy learning often suffers from the fact that the training objective is only a surrogate for the true objective of interest (e.g. task success rate), and policy performance can change significantly from epoch to epoch. This makes it difficult to select the best trained model [19, 28, 29]. In our study, we evaluate each policy checkpoint online in the environment in simulation, and report the best policy success rate per training run. We use these ground-truth values to understand the effectiveness of different selection criteria, and confirm that offline policy selection is an important problem, especially in real-world scenarios where large-scale empirical evaluation is difficult.

**(C5) High Sensitivity to Agent Design Decisions.** Prior studies on machine-generated datasets have shown that offline policy learning can be extremely sensitive to hyperparameter choices [19, 28]. In our study, we explore how agent design decisions affect policy performances, including the choice of agent architecture, agent observation space, and hyperparameter choices per algorithm. This results in several practical conclusions that should prove useful to researchers and practitioners alike. We further show that important design decisions made through our study in simulation directly translate to effective policy learning on real world tasks and datasets.

## 3 Study Design

### 3.1 Tasks

We conducted our study across 5 simulated and 3 real world tasks. The tasks were chosen to test a broad range of manipulation capabilities. See Fig 1 and Appendix E for more details.

**Lift (sim + real)).** The robot arm must lift a small cube. This is the simplest task.

**Can (sim + real).** The robot must place a coke can from a large bin into a smaller target bin. Slightly more challenging than Lift, since picking the can is harder than picking the cube, and the can must also be placed into the bin.

**Square (sim).** The robot must pick a square nut and place it on a rod. Substantially more difficult than Lift and Pick Place Can due to the precision needed to pick up the nut and insert it on the rod.

| Dataset | BC | BC-RNN | BCQ | CQL | HBC | IRIS |
|---|---|---|---|---|---|---|
| Can-Worse | 56.7±2.5 | **92.0±1.6** | 29.3±10.9 | 4.0±3.3 | 78.7±3.4 | 77.3±1.9 |
| Can-Okay | 72.0±2.8 | **95.3±1.9** | 58.0±8.6 | 22.0±4.3 | 97.3±0.9 | 96.0±0.0 |
| Can-Better | 83.3±2.5 | **99.3±0.9** | 62.0±5.9 | 20.7±7.4 | 96.7±0.9 | 96.0±0.0 |
| Can-Worse-Okay | 74.7±5.7 | **98.7±1.9** | 50.7±3.8 | 18.7±2.5 | 88.0±1.6 | 87.3±1.9 |
| Can-Worse-Better | 76.0±4.3 | **100.0±0.0** | 48.0±4.9 | 20.7±5.7 | 90.0±1.6 | 91.3±2.5 |
| Can-Okay-Better | 90.7±1.9 | **100.0±0.0** | 68.7±2.5 | 30.7±7.7 | 99.3±0.9 | 98.0±1.6 |
| Square-Worse | 22.0±4.3 | 39.3±3.8 | 5.3±1.9 | 0.0±0.0 | **44.7±6.8** | 38.7±0.9 |
| Square-Okay | 27.3±3.4 | 45.3±2.5 | 6.7±1.9 | 0.0±0.0 | **52.0±2.8** | 42.0±3.3 |
| Square-Better | 58.7±2.5 | **66.0±2.8** | 32.0±4.3 | 0.7±0.9 | 61.3±1.9 | 60.0±1.6 |
| Square-Worse-Okay | 28.7±2.5 | **55.3±0.9** | 8.7±1.9 | 2.7±1.9 | 50.7±4.1 | 43.3±2.5 |
| Square-Worse-Better | 46.7±5.7 | **73.3±6.2** | 15.3±2.5 | 1.3±0.9 | 65.3±3.4 | 56.7±3.4 |
| Square-Okay-Better | 56.7±4.1 | **74.0±2.8** | 22.0±4.3 | 1.3±0.9 | 63.3±4.1 | 56.7±3.8 |
| Can-Paired | 64.0±9.1 | 70.0±4.3 | 44.7±1.9 | 6.0±1.6 | 70.7±5.2 | **75.3±1.9** |

Table 2: **Results on Suboptimal Human Data.** We present success rates averaged over 3 seeds for each method across different subsets of the Multi-Human datasets, corresponding to mixtures of demonstrations from "Better", "Adequate", and "Worse" human operators, and finally on a diagnostic dataset with paired success and failure human trajectories for each starting initialization. Results indicate that BC-RNN is a strong baseline, and that Batch RL methods perform poorly across all datasets, even on the simple diagnostic dataset.

**Transport (sim).** Two robot arms must transfer a hammer from a closed container on a shelf to a target bin on another shelf. One robot arm must retrieve the hammer from the container, while the other arm must clear the target bin by moving a piece of trash to the nearby receptacle. Finally, one arm must hand the hammer over to the other, which must place the hammer in the target bin.

**Tool Hang (sim + real).** A robot arm must assemble a frame consisting of a base piece and hook piece by inserting the hook into the base, and hang a wrench on the hook. This is the most difficult task due to the multiple stages that each require precise, and dexterous, rotation-heavy movements.

## 3.2 Data Collection

To study the effect of dataset source, we collected data from three different sources – Machine-Generated, Proficient-Human, and Multi-Human (more details in Appendix B).

**Machine-Generated (MG).** We collected these datasets by first training a state-of-the-art RL algorithm [30] on the Lift and Can task, taking agent checkpoints that are saved regularly during training, and collecting 300 rollout trajectories from each checkpoint. Consequently, these datasets are comprised of mixtures of expert and suboptimal data, and resemble datasets from common offline RL benchmarks [18, 19]. We excluded other tasks because they could not be solved by the RL algorithm even with substantial tuning. See the appendix for more details.

**Proficient-Human (PH) and Multi-Human (MH).** Datasets are collected by humans through Robo-Turk [15, 17], a remote teleoperation platform. The PH datasets consist of 200 demonstrations collected by a single, experienced teleoperator, while the MH datasets consist of 300 demonstrations, collected by 6 teleoperators of varying proficiency, each of which provided 50 demonstrations. The 6 teleoperators consisted of a "better" group of 2 experienced operators, an "okay" group of 2 adequate operators, and a "worse" group of 2 inexperienced operators. These data subsets in the Multi-Human data allowed us to investigate the ability of algorithms to deal with mixed quality human data.

**Observation Modalities.** To study the effect of observation modalities, we capture a diverse set of sensor streams when collecting the dataset, including end-effector, gripper fingers, and joints, ground-truth object poses, and images from an external camera and wrist-mounted camera per robot arm (see Appendix E). We have two observation spaces – "low-dim" and "image". Both include end-effector poses and gripper finger positions, and only differ in whether ground-truth object information is used (low-dim) or whether that information is replaced by the available camera observations (image).

## 3.3 Training and Evaluation Protocols

There are several approaches to offline imitation learning [31–34, 9, 10, 25, 35, 36] and offline reinforcement learning [26, 27, 37–43] (see Appendix A for more discussion on related work). We chose to evaluate 6 algorithms in this study – Behavioral Cloning (BC), BC with an RNN policy (BC-RNN), Hierarchical Behavioral Cloning (HBC) [10], Batch-Constrained Q-Learning (BCQ) [26], Conservative Q-Learning (CQL) [27], and IRIS [20]. BC-RNN, HBC, and IRIS have all been used

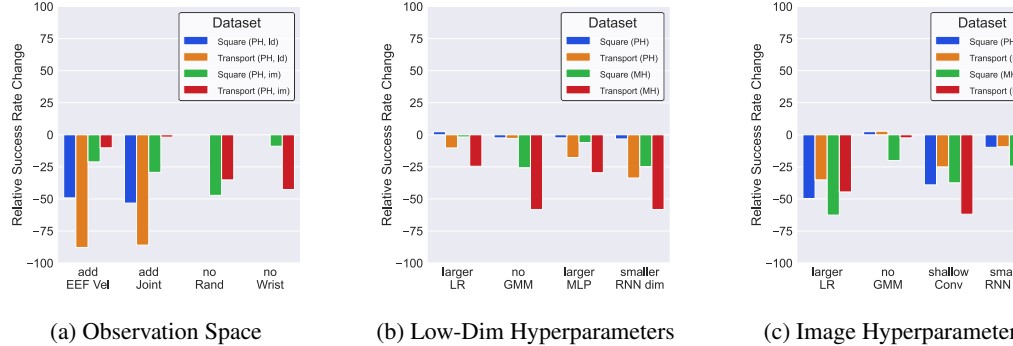

|     |     |     |
| --- | --- | --- |
| (a) Observation Space | (b) Low-Dim Hyperparameters | (c) Image Hyperparameters |

Figure 2: **Effect of Observation Space and Hyperparameter Choice.** We show how the success rate that BC-RNN obtains can drop drastically due to changes to the observation space and hyperparameter settings.

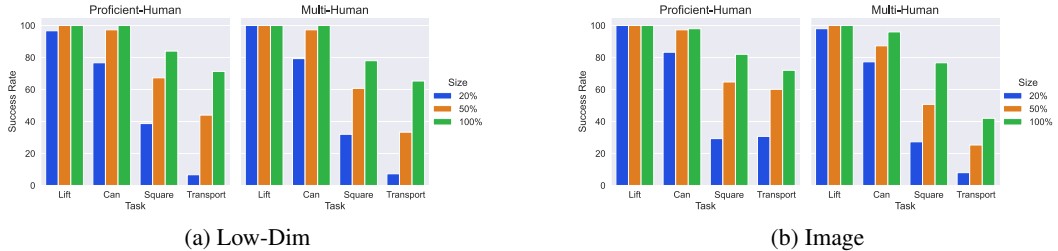

|     |     |
| --- | --- |
| (a) Low-Dim | (b) Image |

Figure 3: **Effect of Dataset Size.** We study how the BC-RNN success rate changes when lowering the quantity of data to 20% and 50%. Results show that less complex tasks (Lift, Can) be learned with a fraction of the data, while more complex tasks might benefit from even larger human datasets.

in prior work to learn offline from teleoperated human demonstrations, while BCQ and CQL are commonly-used offline RL algorithms (see Appendix C). We use binary task completion rewards for all our experiments. Each agent is trained for $N$ epochs, where each epoch consists of $M$ gradient steps, and evaluated every $E$ epochs, by running 50 rollouts in the environment and reporting the success rate over a maximum horizon. For each agent, we report the maximum success rate over the coarse of training, and average over 3 seeds. For low-dim agents, $N = 2000$, $M = 100$, and $E = 50$, and for image agents, $N = 600$, $M = 500$, and $E = 20$ (see Appendix B.2).

## 4 Experiments

In this section, we present each factor that we explored in our study, and note the relevant challenges from Sec. 2 that each pertains to.

### 4.1 Algorithm Comparison on Single and Multi-Human Demonstrations (C1, C2)

We trained and evaluated all algorithms on the Proficient-Human (PH) and Multi-Human (MH) datasets and report the average success rates across 3 seeds in Table 1.

**Observation history is crucial for good performance.** There is a substantial performance gap between BC-RNN and BC, which highlights the benefits of history-dependence. The performance gap is larger for longer-horizon tasks (e.g. $\sim 55\%$ for Transport (PH) compared to $\sim 5\%$ for Square (PH)) and for multi-human data compared to single-human data (e.g. $\sim 25\%$ for Square (MH) compared to $\sim 5\%$ for Square (PH)). Interestingly, results are lower for MH datasets compared to PH datasets, even though the MH datasets contain 100 more demos (300 demos vs. 200 demos). This most likely stems from the presence of suboptimal and multimodal data in the MH datasets.

**Batch RL algorithms perform poorly on Human Datasets.** Recent batch (offline) RL algorithms such as BCQ and CQL have demonstrated excellent results in learning from suboptimal and multi-modal agent-generated datasets. Our results confirm the capacity of such algorithms to work well – BCQ in particular performs strongly on our agent-generated MG datasets that consist of a diverse mixture of good and poor policies. Surprisingly though, neither BCQ nor CQL performs particularly well on these human-generated datasets. This puts the ability of such algorithms to learn from more

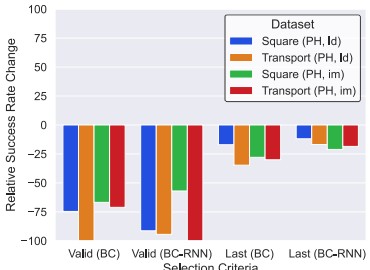

(a) **Effect of Policy Selection Criteria**

Table 3: **Results on Image Observations.**

| Dataset | BC | BC-RNN | BCQ | CQL |
|---|---|---|---|---|
| Lift (PH) | **100.0 ± 0.0** | **100.0 ± 0.0** | 98.0 ± 1.6 | 52.0 ± 13.0 |
| Can (PH) | **97.3 ± 1.9** | 98.0 ± 0.9 | 86.7 ± 2.5 | 0.7 ± 0.9 |
| Square (PH) | 62.0 ± 4.9 | **82.0 ± 0.0** | 41.3 ± 4.1 | - |
| Transport (PH) | 55.3 ± 6.2 | **72.0 ± 4.3** | 0.7 ± 0.9 | - |
| Tool Hang (PH) | 20.0 ± 5.9 | **67.3 ± 4.1** | 3.3 ± 0.9 | - |
| Lift (MH) | **100.0 ± 0.0** | **100.0 ± 0.0** | 93.3 ± 0.9 | 11.3 ± 9.3 |
| Can (MH) | 85.3 ± 0.9 | **96.0 ± 1.6** | 77.3 ± 6.8 | 0.0 ± 0.0 |
| Square (MH) | 46.0 ± 1.6 | **76.7 ± 3.4** | 17.3 ± 7.5 | - |
| Transport (MH) | 18.7 ± 2.5 | **42.0 ± 1.6** | 0.0 ± 0.0 | - |

Figure 5: (left) **Effect of Policy Selection Criteria.** We compare how performance decreases when choosing the policy to evaluate by using the lowest validation loss, or when using the final trained checkpoint, with respect to the best policy performance. (right) **Results on Image Observations.** We present success rates for each method across the image observation human datasets. BC-RNN maintains nearly the same performance as learning from ground-truth observations, providing an optimistic view for learning with real-world raw sensory observations.

natural dataset distributions into question (instead of those collected via RL exploration or pre-trained agents). There is an opportunity for future work in batch RL to resolve this gap.

## 4.2 Learning from Suboptimal Human Data (C2)

To further investigate how algorithms deal with suboptimal human data, we split our MH datasets into smaller subsets based on the proficiency of the human operators. The MH-Better, MH-Okay, and MH-Worse are the 100 demo subsets corresponding to the 2 "better", 2 "okay", and 2 "worse" operators respectively, while MH Worse-Okay, MH Worse-Better, and MH Okay-Better are the 200 demo subsets corresponding to the mixture of the previous subsets. Similar data mixtures have been used for evaluations in batch RL [18]. Appendix B shows the average trajectory lengths in each data subset – lower quality datasets contain demonstrations that take more time to solve the task.

**BC-RNN is a strong baseline on suboptimal human data, but there is room for improvement.** Table 2 shows that BC exhibits a large performance gap between the Better and Worse 100-demo subsets (roughly 27% and 35% for Can and Square respectively). Interestingly, BC-RNN is able to nearly eliminate this gap in performance on the Can task, but not on the Square task. However, BC-RNN outperforms BC on all datasets (7%-35% improvement). Comparing results on the 100 Better demonstrations and 100 Okay demonstrations to the 200 Worse-Better demonstrations and 200 Worse-Okay demonstrations further allows us to analyze how adding 100 "worse" demonstrations impacts the performance of each algorithm. Most algorithms decline in performance while BC-RNN is able to uniformly improve from the added data. Comparing the performance of BC-RNN on the 200-demo Square mixture datasets (55.3%, 73.3%, 74.0%) to the high-quality 200-demo Square (PH) dataset (84.0%) shows that there is still room for algorithms to improve on the use of this data.

**Diagnostic dataset shows that Batch RL struggles in simpler settings as well.** The final row of Table 2 shows additional results on a diagnostic dataset termed Can-Paired, where a single operator collected 2 demonstrations for each of 100 task initializations – one successful demonstration, and one where the can is tossed outside of the bin (task failure), for a total of 200 demonstrations. There is a strong expectation for batch RL algorithms to be able to distinguish between actions leading to successful placement and actions leading to task failure, but even in this simple setting, most algorithms suffer, providing a pessimistic view of the state-of-the-art. The 5% improvement that IRIS provides over BC-RNN suggests that introducing history-dependence into state-of-the-art batch RL algorithms might be a promising direction for future work.

## 4.3 Effect of Observation Space (C5)

**Learning from image observations can match low-dim agent performance.** In Table 3, we present policy learning results when using image observations instead of ground-truth object locations – an important setting for real-world policy learning. BC-RNN still maintains superior performance improvements over BC on the complex Square and Transport tasks, and with the exception of Transport (MH), maintains nearly the same performance as learning from ground-truth observations. This result provides an optimistic view for learning with real-world raw sensory observations.

**Features used for robot proprioception can matter.** In Fig 2a, we study the effect of adding end effector velocities to the observations (+ EEF Vel), and joint positions and velocities to the observations (+ Joint). Surprisingly, we find that including end effector velocity information, and joint information hurts agents trained on low-dim observations substantially (49%-88% relative performance drop), while image-based agents are more tolerant to the inclusion of this extra information (2%-29% relative performance drop). We hypothesize that performance drops might be due to overfitting to the presence of this extra information not needed for solving these tasks. Thus, practitioners should take care to engineer the robot observation space and exclude possibly irrelevant information – information-hiding can be a powerful paradigm for training proficient robots [44].

**Image randomization and wrist observations can be crucial for manipulation tasks.** In Fig 2a, we report performance drops from removing pixel shift image randomization (- Rand) and the wrist camera (- Wrist) from image-based agents to understand their importance. We see that removing randomization results in 47% and 35% relative performance drops on Square and Transport respectively, and removing wrist images results in 9% and 43% relative drops. Consequently, both wrist camera images and image randomization play a substantial role in producing performant policies. We confirm the importance of each for visuomotor imitation in the real world as well (see Sec 4.7). Wrist observations likely help the robot improve gripper alignment for grasping and randomization helps the policy develop invariance for portions of the image that are not important for action prediction.

## 4.4 Effect of Hyperparameter Choice (C5)

In this section, we take our default hyperparameters for BC-RNN and study the effect of changing a subset of them to report practical recommendations for learning from human datasets (see Appendix I for BCQ and CQL). We present our results in Fig 2b (low-dim) and Fig 2c (image).

**(larger LR)** Increasing the learning rate from 1e-4 to 1e-3 affects the performance of image-agents substantially (drop of 35%-63%), while low-dim agents are more tolerant to the change. **(no GMM)** Using a deterministic policy instead of learning a GMM action distribution results in significant relative performance drops on the MH datasets (especially low-dim Transport, with a drop of 58%). **(larger MLP)** Using a larger MLP size at each RNN timestep reduces performance uniformly, suggesting that it is possible to overfit to dataset actions if network architectures are too large. **(shallow Conv)** Using a shallow convolutional network [45] instead of the ResNet backbone [46] for encoding image observations reduces performance significantly – with relative drops of 25%-62%, suggesting that large-capacity visual encoders are crucial for visuomotor imitation. **(smaller RNN dim)** Reducing the size of the RNN hidden dimension from 400 to 100 (low-dim) and 1000 to 400 (image) uniformly decreases performance (drops of 3%-58%), showing the importance of a large RNN hidden dimension. **(Recommendations)** We recommend tuning the LR (especially for image agents) and network structure (MLP size, size of RNN dim) carefully. Opting to use a GMM policy and a ResNet encoder appears to be uniformly better.

## 4.5 Selecting a Policy to Evaluate (C4)

Model selection in offline policy learning can be challenging – for this reason, in our simulation experiments, we evaluated every policy checkpoint online and reported the best one. This is not feasible for real-world settings, making offline policy selection desirable. In Fig 4a, we show that this can be non-trivial, by showing the relative performance drop when selecting the policy using the best loss on validation data (common in supervised learning), and when using the final training checkpoint as well (common in offline RL [47, 18, 27]) – in both cases, the selected policy is significantly worse than the best one (10% to 100% decrease). See Appendix G for more detailed results and discussion. This motivates the need for better offline evaluation metrics.

## 4.6 Effect of Dataset Size (C3)

To study how dataset size impacts performance, we formed smaller 20% and 50% subsets of our human datasets by sampling trajectories. We evaluate low-dim and image BC-RNN agents across these subsets in Table 27 and Table 28. There are several promising results here. We first note that less complex tasks (Lift, Can) can yield proficient policies (75%-100% success rate) using a small fraction of the data (20%). Second, while policies trained on more complex tasks (Square, Transport) suffer substantially when using 50% or 20% of the data, the converse is also true – adding more data

(e.g. moving from 20% to 50% or 50% to 100% size) can result in significant policy improvement. This confirms the value of using large human datasets as a means to obtain proficient policies for challenging and complex manipulation tasks.

### 4.7 Applicability to Real-World Settings

Here, we show that design decisions made in simulation can potentially transfer to real world settings. We collected 3 additional real-world datasets with a Franka robotic arm – Lift (Real), Can (Real), and Tool Hang (Real). Each consists of 200 trajectories collected by one operator. We train BC-RNN and report the final policy checkpoint success rate, over 30 rollouts, due to the time-consuming nature of real world policy evaluation. We also emphasize that no real-world hyperparameter tuning took place, so our results are a lower bound. We were able to train proficient **Lift (96.7%)** and **Can (73.3%)** policies, and the **Tool Hang (3.3%)** policy is able to generate some task successes, despite the extremely difficult nature of the task. Furthermore, as in Sec. 4.3, we validate the importance of pixel shift randomization and the wrist camera by ablating each component on the Can task, and show that including both is the difference between a proficient and non-proficient real-world policy – **Can (- Rand) (26.7%)**, **Can (- Wrist) (43.3%)**.

## 5 Discussion

In this section, we summarize the lessons from our study and make recommendations for future work.

**(L1) Models with temporal abstraction can be extremely effective in learning from human datasets.** In Sec 4.1 and Sec 4.2, we demonstrated that history-dependent models (BC-RNN, HBC, and IRIS) are particularly effective in learning from human datasets compared to algorithms that do not take temporal context into account.

**(L2) Need to improve the ability of batch (offline) RL to learn from suboptimal human datasets.** Sec 4.2 and Appendix I demonstrated that state-of-the-art batch RL algorithms are excellent at learning from suboptimal machine-generated datasets but much worse at learning from suboptimal human datasets. They even struggled with a diagnostic dataset with paired good and bad human demonstration trajectories while IRIS was able to improve slightly on BC-RNN, suggesting that combining history-dependence with value learning might be a good place to start for improving batch RL methods [48–51]. This also demonstrates a need to start benchmarking new batch RL algorithms on human datasets instead of purely on machine-generated datasets.

**(L3) Improving offline policy selection is important for real world settings.** Sec 4.5 demonstrated the need for better ways to select an evaluation policy in an offline manner. We hope that our datasets can help supplement other efforts [29].

**(L4) Observation space plays a large role and hyperparameters matter.** Sec 4.3 demonstrates that policies trained on low-dim observations can be very sensitive to the choice of robot proprioception, while pixel shift randomization and wrist camera images are critical for effective visuomotor policy learning. The choice of observation space for imitation merits careful consideration – other work has also confirmed the importance of feature representations used for offline policy learning [52, 53]. Sec 4.4 and Appendix I made practical recommendations for choosing hyperparameters to learn from human data.

**(L5) There is substantial promise for solving more complex tasks using large-scale human datasets.** Sec 4.6 showed that adding more data can result in significant policy improvement on complex tasks. Table 3 and Sec 4.7 shows that we could learn proficient policies on the Tool Hang task, our most complex task, without any hyperparameter tuning on the task or dataset. Together, these results show the potential of large human datasets as a means to solve challenging and complex manipulation tasks.

**(L6) Study results transfer to real-world settings.** In Sec 4.7, we showed that we could directly apply hyperparameters that were tuned on simulated tasks directly to real-world datasets and tasks. This provides promise for using our tasks, datasets, and codebase to enable reproducible evaluation in simulation, while also being confident that conclusions can transfer to real-world settings.

Going forward, we hope that the datasets, tasks, code, and subsequent insights of our study will serve researchers and practitioners alike.

**Acknowledgments**

We would like to thank Albert Tung for helping with the RoboTurk data collection system, Jim Fan for providing timely lab cluster support, and Helen Roman for helping order items for the physical robot tasks. Ajay Mandlekar acknowledges the support of the Department of Defense (DoD) through the NDSEG program. We acknowledge the support of Toyota Research Institute ("TRI"); this article solely reflects the opinions and conclusions of its authors and not TRI or any other Toyota entity. We acknowledge the support of the US Army Research Office (award W911NF-15-1-0479) and the National Science Foundation (award CNS-1955523). This work relates to Department of Navy award N00014-14-1-0671 issued by the Office of Naval Research.

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
