# OpenReview forum: "What Matters in Learning from Offline Human Demonstrations for Robot Manipulation"
_robot-learning.org/CoRL/2021/Conference — CoRL2021 Oral_

### Official Review · Reviewer_mRsh · 2021-07-20

**Originality:** Fair
**Technical Quality:** Good
**Clarity Of Presentation:** Very Good
**Impact:** 3

**Recommendation:**

Strong Accept: I recommend accepting the paper and will argue for my recommendation even if other reviewers hold a different opinion.

**Summary:**

The paper conducts a study of evaluating several imitation learning and batch offline RL algorithms when trained on offline demonstrations in simulated and real-world environments. In particular, such algorithms as Behavioral Cloning (BC), BC with recurrent neural networks, Hierarchical BC, Batch-Constrained Q-Learning (BCQ), Conservative Q-Learning (CQL) and IRIS are evaluated. Experiments include simulated and real lifting, simulated and real pick-and-place, simulated high-precision pick-and-place, simulated bi-manual transport, simulated and real articulate tool hanging tasks. Datasets used for experiments comprise machine-generated, and human-generated data, whereas human demonstrations are categorized based on the level of experience of the demonstrator. Overall the main takeaways from experiments are that using recurrent models is important for learning from human demonstrations, batch offline RL algorithms do not work well with human-generated data, offline policy selection is important ant it is difficult in the real world, including certain observation dimensions can lead to increased overfitting and more data leads to increased performance. Paper also provides some practical ways to tune hyper-parameters, and demonstrates that using algorithms and parameters selected in simulation transfers to learning on a real robot.

**Issues:**

Please refer to "Strengths And Weaknesses" for a detailed list of issues.

**Reviewer Expertise:**

Very good: Comprehensive knowledge of the area

**Strengths And Weaknesses:**

Strengths:
- The problem of learning from offline demonstration data is important for initializing robot policies, improving their safety and scaling up robot learning.
- The paper is well-written and easy to understand and follow.
- The paper conducts a wide range of experiments, including real-world robot evaluations on non-trivial tasks.
- The paper contains a range of practical insights that will be useful for robot learning practitioners.

Weaknesses:
- In general, although offering a great comparison of various algorithms, the paper doesn’t introduce novel algorithmic components.
- It is not always completely clear how true rewards are used during the optimization. For example, machine-generated and can-paired data contain both, success and failures. It looks like BC-based algorithms are not using the success/failure label and batch RL algorithms are using them. It would be useful if paper stated more clearly how these ground truth labels are used.
- The paper makes a conclusion that batch RL algorithms do not work well with human-generated data. However, it is indicated that for machine-generated data, 6 checkpoints with 300 rollouts each are included in the dataset, whereas human-generated datasets only include 200 or 300 demonstrations in total, so at least 6 times less data. Another conclusion that can be made from these experiments is that batch RL algorithms work worse with a low amount of data. It would be interesting to see experiments that disambiguate these two conclusions.

**Summary Of Recommendation:**

The paper provides a great comparison of various algorithms for offline imitation learning, but lacks some novelty in theoretical contributions.

---

> ### Author Response · Authors · 2021-08-21
> **Response to Reviewer mRsh**
>
> Please note that our response has been split into two parts due to space constraints. (Part 1/2)
>
> Thank you for your thorough and helpful comments! We are happy to hear that you found our study to contain a wide range of experiments and several practical and useful insights. You brought up some great questions and suggestions that we discuss further below. Please let us know if further clarification is needed.
>
> - “In general, although offering a great comparison of various algorithms, the paper doesn’t introduce novel algorithmic components.”
>
> Our intention with this study was to highlight the challenges that current algorithms face in learning from human datasets, and to develop several insights to guide future work. However, there is tremendous value in developing new theories and new algorithms using the insights presented in this study. We hope that the release of our code and datasets will make it easy for future work to reproduce our work, build on our study, and develop new algorithms.
>
> Furthermore, we feel that the extensive empirical evaluation that went into developing and substantiating the lessons in the study is important for the community, even if the lessons themselves may not be very novel or surprising to those familiar with this setting. While the general trends may seem obvious to experts in the research field, we believe that the concrete evidence in terms of quantitative evaluation, the magnitude of the performance differences behind algorithms, and design decisions that the paper explores is highly relevant for the community and important for guiding future research. We list a few examples below.
>
> 1. It may seem intuitive that models with history-dependence, such as BC-RNN should outperform models with no history-dependence like BC, since humans might make decisions that depend on the past. However, the performance gap can be extreme (e.g. Table 1, Transport PH, there is a 54% gap in success rate, 71.3% for BC-RNN vs. 17.3% for BC).
>
>
> 2.  Offline RL algorithms like BCQ perform worse than BC-RNN on human datasets - this is a general qualitative trend. However, once again the gap can be extreme (e.g. Table 1, Transport PH, there is a 64% gap in success rate). We also found this conclusion to be somewhat surprising, given that recent research has employed offline RL as an effective tool to learn from large mixed quality datasets.
>
>
> 3. The fact that action loss for behavioral cloning corresponds poorly to the task success rate of a policy is well known, but how poor this correspondence is in practice has not been extensively studied, while we present concrete results in Figure 5a. This motivates a need for more sophisticated mechanisms for offline policy selection.
>
>
> 4. The fact that offline learning algorithms are sensitive to hyperparameter choices and observation spaces is also well known, but the degree of the dependence is something that we study extensively (see Figure 2). We saw that careful robot proprioception design is critical, and including too much information can cut performance by over 80%.
>
>
> 5. Another surprising result is that the images provided to the agent for visuomotor imitation also require careful consideration - including wrist camera observations and performing pixel shift randomization is crucial, and without it, policies can drop in performance by up to 50%. This is the difference between policies that work and policies that don’t.
>
>
> 6. It’s also well known that performance should increase with dataset size - but how does it vary by task complexity, and what is the shape of the curve (e.g. is there linear dependence)? The results in Figure 3 are a starting point for answering such inquiries.
>
> Furthermore, we think that having these lessons explicitly stated in one location along with evidence to back up the claims would be useful to those not familiar with the setting of learning from human demonstrations, and help serve as an entrypoint for them.
>
>
> - “It is not always completely clear how true rewards are used during the optimization...It would be useful if paper stated more clearly how these ground truth labels are used.”
>
> We apologize for the confusion and have attempted to make this more clear in the revised text. We have added text in section 3.3, and more explicit clarification in Appendix C on the form of the reward and done signals used by the algorithms we evaluated (the added text is highlighted in red in the revised submission). We also summarize the content here - the reward and done signals are only used by batch RL algorithms (BCQ, CQL, and IRIS). We use binary task completion rewards for all of our experiments (R(s, a, s’) = 1 if the task is solved in state s’). Similarly, the done signal for a transition (s, a, s’) is true if the task is solved in state s’, or if it is the last transition in the trajectory.

---

> > ### Author Response · Authors · 2021-08-21
> > **Response (Part 2)**
> >
> > Please note that our response has been split into two parts due to space constraints. (Part 2/2)
> >
> > - “Another conclusion that can be made from these experiments is that batch RL algorithms work worse with a low amount of data. It would be interesting to see experiments that disambiguate these two conclusions.”
> >
> > We agree that the scale of the machine-generated data is higher than that of the human data, and that a completely valid conclusion from the current experiments is that batch RL algorithms perform worse with lower amounts of data. And we appreciate your suggestion to conduct experiments that disambiguate the two conclusions. We took our machine-generated lift and can dataset, and curated a subset by using 200 demonstrations from the first agent checkpoint (effectively random) and 100 demonstrations from the last agent checkpoint (a proficient agent) to form a 300 demonstration subset, in order to match the number of demonstrations in the multi-human datasets. We then trained both BC and BCQ on this subset, with 3 seeds each. The results are as follows - on the Lift subset, BC achieves 84.7+/-2.49 and BCQ achieves 90.0+/-1.63, and on the Can subset, BC achieves 64.7+/-6.80 and BCQ achieves 73.3+/-2.49. This shows that batch RL algorithms can learn from smaller machine-generated datasets.
> >
> > Additionally, in our study, imitation learning algorithms like BC-RNN and HBC show that it is possible to do quite well on human datasets despite only containing 200-300 trajectories. It would be interesting to collect larger human datasets to see if this helps lower the performance gap between imitation and batch RL algorithms -- we plan to study this in the future.

---

> > > ### Comment · Reviewer_mRsh · 2021-08-23
> > > **Thanks for addressing my comments**
> > >
> > > Thank you for addressing my comments in detail. Most of my concerns have been addressed and I have increased the rating.

---

### Official Review · Reviewer_wVDW · 2021-07-23

**Originality:** Good
**Technical Quality:** Very Good
**Clarity Of Presentation:** Very Good
**Impact:** 3

**Recommendation:**

Weak Accept: I recommend accepting the paper, but will not argue for my recommendation if the majority of other reviewers have a different opinion.

**Summary:**


This paper conducts extensive evaluations on learning from offline demonstrations for robot manipulation tasks. Specifically, it studies six offline learning algorithms (i.e. BC, BC-RNN, HBC, BCQ, CQL, and IRIS) on five simulated and three real-world robot manipulation tasks of varying complexity with datasets of varying quality. This work presents a variety of lessons learned from the study and makes recommendations for future work accordingly. In sum, this work presents systematical analyses of existing methods focused on robot manipulation with offline demonstrations. I believe this kind of work benchmarking existing methods is one key to measure the advance of the field. While I have some concerns regarding limited robot configuration and demonstrations, I am leaning toward accepting this paper. Yet, my research interests are not directly related to this research direction, and therefore my evaluations should be taken lightly.

**Issues:**


Described in the strengths and weaknesses section.

**Reviewer Expertise:**

Fair: Some knowledge of the area

**Strengths And Weaknesses:**


## Paper strengths and contributions
**Motivation**
Most of the recent papers studying robot manipulation with offline datasets usually set up their own environments/tasks and collect their own demonstrations. This makes it very difficult to draw a fair comparison among the methods proposed in these papers or measure the advance in this field. Therefore, I believe systematically benchmark existing methods would be beneficial to the community.

**Interesting insights**
This work presents a variety of insights on learning robot manipulation tasks with offline demonstrations. The following findings are particularly interesting to me:
- Leveraging history is important when learning human demonstrations based on the performance gap between BC and BC-RNN.
- Offline RL methods perform well when learning from agent-generated demonstrations but not from human demonstrations
- Including end effector velocity and joint information hurts agents trained on low-dim observations
- Selecting the policy using the best loss on validation data or using the final training checkpoint yields a significant performance drop compared to the real best policies.

**Clarity**
- The overall writing is clear, including what the goals are, how the experiments are set up and conducted, etc.
- The challenges are clearly stated.
- The findings are well-organized.
- The presentation of the experimental results is clear.
- Most of the analyses of the experimental results are detailed, comprehensive, and convincing.

**Environments & tasks & demonstrations**
The effort on setting put the environments in simulation and real-world as well as collecting demonstrations is highly appreciated.

**Diverse demonstrations**
- Evaluating methods learning from demonstrations of varying quality provides informative findings.
- Comparing models learning from agent-produced demonstrations and human-generated demonstrations presents interesting insights.
- Training models to learn from different numbers of demonstrations yields results that align with intuitions.

## Paper weaknesses and questions

**Limited robot configuration**
This work only conducts studies with a Franka robotic arm in both simulated and real-world environments. Therefore, it is not easy to judge if the majority of the findings presented in this work would apply to other robot configurations. While I do recognize the difficulty of accessing a variety of robot configurations, especially for an academic lab, I believe it is still possible to conduct studies with other type of robots at least in simulation to justify the findings.

**Converged machine-generated demonstrations**
The machine-generated dataset consists of demonstrations collected from checkpoints along with the training of expert policies. It makes sense to do so to collect suboptimal demonstrations and it may seem fair since human-generated datasets probably also contain trajectories collected when human demonstrators just start learning to use the teleoperation platform. However, human demonstrators do have an understanding of the tasks from the start and have prior knowledge such as physics. Therefore, I believe the suboptimal trajectories from the machine-generated datasets could be much worse than the suboptimal ones in the human-generated datasets, which can even enlarge the performance gap between models learning from the machine and human-generated demonstrations.

**Short-horizon tasks**
This paper mainly studies robot manipulation tasks with short horizons. While tasks such as Tool Hang and Transport consisting of multiple subtasks and might take hundreds of time steps, they do not examine robots' ability to plan. Therefore, the evaluations are all about the precision of the execution but not about the ability to interpret the intentions or the high-level plans of demonstrators.

**Domain gap**
The studies presented in this work only concern the setup where the agent and demonstrators share exactly the same environment. Yet, in many cases of real-world applications, robots might need to deal with subtle differences between the environment it faces and the one from the demonstration. I believe it would be also important to study this case where this domain gap exists.

**Summary Of Recommendation:**


This work presents systematical analyses of existing methods focused on robot manipulation with offline demonstrations. I believe this kind of work benchmarking existing methods is one key to measure the advance of the field. While I have some concerns regarding limited robot configuration and demonstrations, I am leaning toward accepting this paper. Yet, my research interests are not directly related to this research direction, and therefore my evaluations should be taken lightly.

---

> ### Author Response · Authors · 2021-08-21
> **Response to Reviewer wVDW**
>
> Please note that our response has been split into two parts due to space constraints. (Part 1/2)
>
> Thank you for your thorough and helpful comments! We are happy to hear that you found our study to be comprehensive and convincing and that you agree that our insights are interesting. You brought up some great questions and suggestions that we discuss further below. Please let us know if further clarification is needed.
>
> - Limited Robot Configuration
>
> This is a great point! We chose to limit the scope of the study in order to feasibly carry out the large number of evaluations necessary for completing the study (5 sim + 3 real tasks, 6 algorithms, 3 dataset sources, 2 observation spaces, and several hyperparameter choices). However, we do indeed have some evidence that the results transfer to other robotic platforms. The IRIS paper (https://arxiv.org/abs/1911.05321) presented results on the crowdsourced RoboTurk cans dataset (https://roboturk.stanford.edu/dataset_sim.html), which consists of a Sawyer robotic arm, and 100 hz joint velocity actions. The authors showed that they could achieve a success rate of 30%, which was the best reported performance from offline training on this dataset (for training on low-dimensional observations). We used our study insights to train a BC-RNN model that achieves over 60% success rate (double the performance!). Note that this dataset contains a different robot (Sawyer instead of Franka) and a different action space (100 hz joint velocity instead of 20 hz operation space end effector control).
>
> Also, our results on the two arm transport task suggest that our insights may transfer to bimanual robot morphologies, where two arms must be controlled simultaneously.
>
> Furthermore, we do not see any theoretical reasons why the general observations we made about learning from human datasets would not extend to similar types of robot arms (in our case, 7 degree-of-freedom arms with a stationary camera observing the scene and another camera attached to the end-effector). Differences in morphology may of course affect the results, but we believe that they would not affect the general conclusions we drew. This would be best supported by further experimental evaluation.
>
> Finally, the robosuite simulation framework, in which we developed our simulation tasks, easily supports replacing the Franka arm with other arms such as the Sawyer, and further supports the collection of human demonstrations via keyboard and 3d mouse interfaces. Our released code will be compatible with human demonstrations collected through the robosuite framework, making it easy for other researchers to explore datasets that involve other robotic arms.
>
>
> - Converged machine-generated demonstrations
>
> We acknowledge that the suboptimal machine-generated trajectories are certainly worse in many cases than the suboptimal human trajectories (especially since many are unsuccessful), but this should bias performance to be lower on machine-generated data. However, we found that batch RL algorithms are proficient on machine-generated data, but perform poorly on human data. For example, on the machine-generated Lift dataset, BCQ significantly outperforms BC-RNN (91.3% vs. 70.7%), but BCQ struggles on human datasets -- on the Transport (PH) dataset, BC-RNN achieves 71.3% task success rate, but BCQ achieves only 7.3%, and on the Can (MH) dataset, BC-RNN achieves 100% while BCQ achieves 62.7% (see Table 1 and Table 2 for more examples).

---

> > ### Author Response · Authors · 2021-08-21
> > **Response (Part 2)**
> >
> > Please note that our response has been split into two parts due to space constraints. (Part 2/2)
> >
> > - Short-horizon tasks
> >
> > While learning to plan for long-horizon tasks from demonstrations is a highly relevant line of research (e.g., [1], [2], and [3]) and an interesting avenue for future work, the focus of this paper is rather on learning closed-loop policies for fine-grained robot manipulation tasks. Furthermore, if you take a look at Table 4 and Table 5 in Appendix B (see supplementary), the task horizons can range from 50 to 1000 timesteps. As you pointed out, the Transport and Tool Hang tasks in particular are multi-stage, and require composing different manipulation behaviors together to solve successfully.
> > [1] Lynch, Corey, et al. "Learning latent plans from play." Conference on Robot Learning. PMLR, 2020.
> > [2] Gupta, Abhishek, et al. “Relay Policy Learning: Solving Long-Horizon Tasks via Imitation and Reinforcement Learning”. Conference on Robot Learning. PMLR, 2020.
> > [3] Xu, Danfei, et al. "Regression planning networks." Proceedings of the 33rd International Conference on Neural Information Processing Systems. 2019.
> >
> >
> > - Domain gap
> >
> > To be clear, the set of tasks considered in the study require trained policies to generalize to an entire start state distribution (see this link for visualization). It is highly likely that trained policies will need to deal with initial states that were not present in the recorded data because the set of initial states in the demonstration data is a strict subset of the possible initial states in the start state distribution. However, we did not study the setting of domain shift, where the train and test start state distributions would be different -- we leave this for future work.

---

> > > ### Comment · Reviewer_wVDW · 2021-08-24
> > > **Re: Response**
> > >
> > > Thanks for the detailed response. My assessment stays the same as It seems that my understanding of the paper is accurate.
> > >
> > > Limited Robot Configuration: "... we do not see any theoretical reasons why the general observations we made about learning from human datasets would not extend to similar types of robot arms": this is not a particularly convincing state since there are also no theoretical reasons the findings should generalize across different types of robots. The findings are empirical and therefore I believe the authors would need to present empirical results to demonstrate that.

---

### Official Review · Reviewer_YubL · 2021-07-23

**Originality:** Good
**Technical Quality:** Very Good
**Clarity Of Presentation:** Excellent
**Impact:** 4

**Recommendation:**

Weak Accept: I recommend accepting the paper, but will not argue for my recommendation if the majority of other reviewers have a different opinion.

**Summary:**

The paper presents an exploratory study aimed at identifying over-arching challenges in learning manipulation skills from offline human-labelled datasets. Specifically, by varying factors such as the learning algorithm, dataset size, optimality of demonstrations, and task complexity, the study compared the performance of policy learned under various permutations of choices. Finally, the work distills six key lessons learned from the study.

**Issues:**

1. Would it be possible to compute more metrics (in addition to success rate) in the future if all the data and code are made available?
2. Given that all the experiments are conducted on a single robot platform (the Franka arm) brief discussion surrounding the generalizability of these finds across different robot morphology would be very helpful.

**Reviewer Expertise:**

Very good: Comprehensive knowledge of the area

**Strengths And Weaknesses:**

**Strengths**

+ The paper presents a timely and important study that explores the challenges involved in learning manipulation tasks from offline demonstrations.
+ The paper is well-written and very easy to read.
+ Making the data and code available would make a big impact in the research communicate and help benchmark and compare new approaches.
+ I really appreciate the careful and comprehensive design of the study that includes variations across various important axes such as task complexity, user expertise, perception modality, and learning paradigm (offline RL and imitation).
+ Almost all design choices are carefully motivated and explained.

**Weaknesses**

- The paper is missing a few key existing works that attempt to benchmark learning from demonstrations (Rana et al., ICRA, 2020; Lemme et al., J. Beh. Robo., 2015) . It is important to acknowledge these contributions and contextualize the contributions made by this work within those of existing studies. While do not believe that these work subsume all contributions of the proposed work, there are non-trivial overlaps.
- It is helpful to see the lessons learned clearly articulated and have numerical evidence to support them. However, the reported lessons are not surprising and seem intuitive.
- The evaluation metrics seems to be limited to success rates. It would be more illuminate to analyze more metrics such as time taken to perform the task, smoothness of motion, and (subjective) user perception.

**References**

- Rana, et al. "Benchmark for Skill Learning from Demonstration: Impact of User Experience, Task Complexity, and Start Configuration on Performance." 2020 IEEE International Conference on Robotics and Automation (ICRA). IEEE, 2020.
- Lemme, Andre, et al. "Open-source benchmarking for learned reaching motion generation in robotics." Paladyn, Journal of Behavioral Robotics 6.1 (2015).

**Summary Of Recommendation:**

On the one hand, the paper presents a timely and comprehensive study of the challenges involved in learning manipulation policies from offline datasets. The study design is commendable and making the datasets and code available to the public will likely have a big impact on the research community. On the other hand, the paper does not acknowledge a couple of key prior studies with similar aspirations, and evaluates the performance of each permutation of factors solely based on task success rate.

---

> ### Author Response · Authors · 2021-08-21
> **Response to Reviewer YubL**
>
> Please note that our response has been split into two parts due to space constraints. (Part 1/2)
>
> Thank you for your insightful comments! We are happy to hear that you found our study to be careful and comprehensive in exploring several important dimensions and that you agree that our data and code release would have a significant impact on the community. You brought up some great suggestions and questions that we discuss further below. Please let us know if further clarification is needed.
>
> - “The paper is missing a few key existing works that attempt to benchmark learning from demonstrations (Rana et al., ICRA, 2020; Lemme et al., J. Beh. Robo., 2015).”
>
> Thank you for pointing out these existing works - we were not aware of them previously. Please take a look at the revised related work section (Appendix A in the supplementary material, with the changes highlighted in red). We now summarize the important differences.
>
> Lemme et al. focuses on learning point-to-point reaching motions from demonstrations while we focus on tabletop manipulation settings where the robot must interact with one or more objects. However, unlike our study, Lemme et al. also evaluates the robustness of the learned robot motions by introducing perturbations. While similar mechanisms could be used to understand the robustness of the policies trained in this study, and such evaluations are important for deploying policies in real-world settings, we leave this for future work.
>
> Similar to our study, Rana et al. collected demonstration trajectories across many humans and tasks. However, their datasets consist of robot end effector trajectories, while our datasets and learning methods focus on leveraging additional modalities such as object poses and camera images to train policies that can solve tasks across several scene configurations. Interestingly, Rana et al. also used crowdsourced humans to evaluate learned robot motions with subjective metrics such as safety, while we primarily evaluate our policies using task success rate. Subjective measures like safety are important for real-world policy deployment, and it will be interesting to study this further in the future.
>
>
> - “Given that all the experiments are conducted on a single robot platform (the Franka arm) brief discussion surrounding the generalizability of these finds across different robot morphology would be very helpful.”
>
> This is a great point! We chose to limit the scope of the study in order to feasibly carry out the large number of evaluations necessary for completing the study (5 sim + 3 real tasks, 6 algorithms, 3 dataset sources, 2 observation spaces, and several hyperparameter choices). However, we do indeed have some evidence that the results transfer to other robotic platforms. The IRIS paper (https://arxiv.org/abs/1911.05321) presented results on the crowdsourced RoboTurk cans dataset (https://roboturk.stanford.edu/dataset_sim.html), which consists of a Sawyer robot arm, and 100 hz joint velocity actions. The authors showed that they could achieve a success rate of 30%, which was the best reported performance from offline training on this dataset (for training on low-dimensional observations). We used our study insights to train a BC-RNN model that achieves over 60% success rate (double the performance!). Note that this dataset contains a different robot (Sawyer instead of Franka) and a different action space (100 hz joint velocity instead of 20 hz operation space end effector control).
>
> Also, our results on the two arm transport task suggest that our insights may transfer to bimanual robot morphologies, where two arms must be controlled simultaneously.
>
> Furthermore, we do not see any theoretical reasons why the general observations we made about learning from human datasets would not extend to similar types of robot arms (in our case, 7 degree-of-freedom arms with a stationary camera observing the scene and another camera attached to the end-effector). Differences in morphology may of course affect the results, but we believe that they would not affect the general conclusions we drew. This would be best supported by further experimental evaluation.
>
> Finally, the robosuite simulation framework, in which we developed our simulation tasks, easily supports replacing the Franka arm with other arms such as the Sawyer, and further supports the collection of human demonstrations via keyboard and 3d mouse interfaces. Our released code will be compatible with human demonstrations collected through the robosuite framework, making it easy for other researchers to explore datasets that involve other robotic arms.

---

> > ### Author Response · Authors · 2021-08-21
> > **Response (Part 2)**
> >
> > Please note that our response has been split into two parts due to space constraints. (Part 2/2)
> >
> > - “The evaluation metrics seem to be limited to success rates. It would be more illuminate to analyze more metrics such as time taken to perform the task, smoothness of motion, and (subjective) user perception”
> >
> > We completely agree that analyzing more characteristics of trained policies would be very interesting. Qualities such as the smoothness of motion, safety consideration, and other subjective impressions based on human evaluations (such as those in Rana et al.) are critical for real-world policy deployment. While this is outside the scope of the current study, our publicly available datasets and code would facilitate such evaluations in future work. We are excited to see how others in the community will use our datasets and code to conduct additional analysis and evaluations.
> >
> >
> > - “Would it be possible to compute more metrics (in addition to success rate) in the future if all the data and code are made available?
> >
> > Yes - absolutely! Users will have complete control over how the data and code is used -- they can reproduce our experiments, train policies for their own purposes, and then evaluate the policies as they see fit.  We also plan to release a model zoo of trained policies that can be loaded and analyzed right out of the box.
> >
> >
> > - “However, the reported lessons are not surprising and seem intuitive.”
> >
> > We feel that the extensive empirical evaluation that went into developing and substantiating the lessons in the study is important for the community, even if the lessons themselves may not be very surprising to those familiar with this setting. While the general trends may seem obvious to experts in the research field, we believe that the concrete evidence in terms of quantitative evaluation, the magnitude of the performance differences behind algorithms, and design decisions that the paper explores is highly relevant for the community and important for guiding future research. We list a few examples below.
> >
> > 1. It may seem intuitive that models with history-dependence, such as BC-RNN should outperform models with no history-dependence like BC, since humans might make decisions that depend on the past. However, the performance gap can be extreme (e.g. Table 1, Transport PH, there is a 54% gap in success rate, 71.3% for BC-RNN vs. 17.3% for BC).
> > 2.  Offline RL algorithms like BCQ perform worse than BC-RNN on human datasets - this is a general qualitative trend. However, once again the gap can be extreme (e.g. Table 1, Transport PH, there is a 64% gap in success rate). We also found this conclusion to be somewhat surprising, given that recent research has employed offline RL as an effective tool to learn from large mixed quality datasets.
> > 3. The fact that action loss for behavioral cloning corresponds poorly to the task success rate of a policy is well known, but how poor this correspondence is in practice has not been extensively studied, while we present concrete results in Figure 5a. This motivates a need for more sophisticated mechanisms for offline policy selection.
> > 4. The fact that offline learning algorithms are sensitive to hyperparameter choices and observation spaces is also well known, but the degree of the dependence is something that we study extensively (see Figure 2). We saw that careful robot proprioception design is critical, and including too much information can cut performance by over 80%.
> > 5. Another surprising result is that the images provided to the agent for visuomotor imitation also require careful consideration - including wrist camera observations and performing pixel shift randomization is crucial, and without it, policies can drop in performance by up to 50%. This is the difference between policies that work and policies that don’t.
> > 6. It’s also well known that performance should increase with dataset size - but how does it vary by task complexity, and what is the shape of the curve (e.g. is there linear dependence)? The results in Figure 3 are a starting point for answering such inquiries.
> >
> > Furthermore, we think that having these lessons explicitly stated in one location along with evidence to back up the claims would be useful to those not familiar with the setting of learning from human demonstrations, and help serve as an entrypoint for them.

---

> > > ### Comment · Reviewer_YubL · 2021-08-22
> > > **I agree whole-heartedly!**
> > >
> > > I now realize that my comment about "unsurprising results" needed more context - I meant to say that the qualitative take-aways are not surprising. In fact, I whole-heartedly agree with the authors' responses. It is indeed important to explicitly state these lessons with quantifiable effects, back them up with evidence, and serve as a reference for those not familiar with LfD.

---

> > ### Comment · Reviewer_YubL · 2021-08-22
> > **Thank you!**
> >
> > I appreciate the authors taking the time to respond to my comments and making appropriate changes. While my overall evaluation of the paper has not changed considerably, I agree with the responses.

---

### Official Review · Reviewer_XeZV · 2021-07-27

**Originality:** Good
**Technical Quality:** Very Good
**Clarity Of Presentation:** Excellent
**Impact:** 4

**Recommendation:**

Strong Accept: I recommend accepting the paper and will argue for my recommendation even if other reviewers hold a different opinion.

**Summary:**

This paper identifies challenges of learning from offline human demonstrations for robot manipulation and analyzes the performance of 3 BC algorithm variants (BC, BC-RNN, HBC) and 3 batch-RL algorithms (BCQ, CQL, and IRIS) across 5 simulated and 3 real-robot manipulation tasks of varying difficulty, 3 different dataset generation sources, two observation spaces, and various hyper-parameter choices, while providing various relevant insights.

**Issues:**

See “suggestions for improvement” section above


**Reviewer Expertise:**

Very good: Comprehensive knowledge of the area

**Strengths And Weaknesses:**

Strengths:
- Identification of challenges of learning from offline human data
- Extensive evaluation across various dimensions such as model choice, observation space choice, dataset source, and hyperparameters for manipulation tasks with varying difficulty
- Useful insights on performance across these variables with suggestions for addressing the challenges

Suggestions for Improvement:

- Can the paper dive a little deep into what differences in the nature of the mixed quality machine-generated datasets when compared to mixed-quality human datasets make it difficult for SOTA RL algorithms such as BCQ to perform comparably with human datasets?

- The effect of observation spaces in the performance for IL algorithms could be more profound than just low-dimension (object state estimation) vs. high-dimension (visual raw data). The observation space is also sensitive to frame of reference (object reference or robot reference) and multi-modality sensing. For example: how would the trends change if force information was available as an observation in addition to the visual data? Or, how would the performance change if the robot states (gripper opening, end-effector poses etc.) are expressed in the object coordinate frame vs. robot-base coordinate frame? Also, what about the transport task made it difficult to be learned using the high-dimensional observation space? And, how does the performance vary when there is uncertainty in the low-dimensional object pose?

- While the paper mentions the promise of using more data for complex tasks with high-dim observation space, collecting huge human datasets is very challenging realistically. Can the paper comment on the feasibility as well?

- The paper can become stronger with comments on how these variables affect the performance in the presence of covariate shift, which BC algorithms generally succumb to

- Can the insights extend beyond robot manipulation problems to other robotic applications? Kindly discuss

- The recommendations given in Section 4.4 (C5 - Hyperparameter choice) are conditioned on a given dataset size. Some of the overfitting concerns with larger models may not exist with larger datasets. Can the paper discuss the recommendations in light of other variables that may affect the performance? I would also revise L6 in the Discussion section by adding caution to the statement that the results transfer to real-world settings, especially because this is conditioned on the difficulty of a task. The paper would benefit by detailing what about a task makes it difficult in this context, is it the long-horizon factor, is it the precision factor, is it multi-arm (agent) factor, or is it all of the above, and are there other aspects of a task that could make it difficult for these learning tasks? These discussions can make the paper much stronger and more useful to the community in general.


**Summary Of Recommendation:**

This is an extensive study and the insights would be valuable to the community. I strongly support accepting the manuscript.

---

> ### Author Response · Authors · 2021-08-21
> **Response to Reviewer XeZV**
>
> Thank you for your incredibly detailed and insightful comments! We are pleased to hear that you found our study to be extensive in exploring several important dimensions and that you agree that our insights would be valuable for the community. You brought up some great questions and suggestions that we discuss further below. Please let us know if further clarification is needed. Please note that our response has been split into two parts due to space constraints. (Part 1/2)
>
>
> - “While the paper mentions the promise of using more data for complex tasks with high-dim observation space, collecting huge human datasets is very challenging realistically. Can the paper comment on the feasibility as well?”
>
> Efforts like RoboTurk (https://arxiv.org/abs/1811.02790, https://arxiv.org/abs/1911.04052) are a great start, and show that it is indeed feasible to collect large datasets from humans, by allowing anyone with a smartphone to collect task demonstrations remotely. We are also happy to see several recent efforts - a number of CoRL submissions this year are also based on collecting large-scale human datasets (https://openreview.net/forum?id=8kbp23tSGYv, https://openreview.net/forum?id=Es1ZD8OOvHO, https://openreview.net/forum?id=TavPBk4Zs9m). We believe that our study highlights the value and importance of understanding the challenges and opportunities in such large-scale human demonstrated datasets and hope that it can inspire the community to dedicate more effort towards the important problem of scaling up data collection from humans and making such datasets prevalent in the community.
>
>
> - “Can the insights extend beyond robot manipulation problems to other robotic applications? Kindly discuss”
>
> We imagine that most challenges that we pointed out such as partial observability in human decision making could appear in other domains such as self-driving cars, since they don’t get to observe all the variables in how the human makes a decision. Sensitivity to hyperparameters and observation space is true for closed-loop policy learning in general, as pointed out by several prior studies (for example, this one https://arxiv.org/abs/1709.06560). We pointed out that offline policy selection is important because trying each and every policy on a real robot can be challenging and time-consuming. This is true for several other domains, and in certain cases, it can also be unsafe to do so (for example, when a policy corresponds to a medical treatment - see this recent work for more discussion https://arxiv.org/abs/2103.16596).
>
>
> - “The recommendations given in Section 4.4 (C5 - Hyperparameter choice) are conditioned on a given dataset size. Some of the overfitting concerns with larger models may not exist with larger datasets. Can the paper discuss the recommendations in light of other variables that may affect the performance?”
>
> We did observe that dataset size is an important factor of learning in this setting - this is what inspired us to conduct the analysis in Section 4.6 and Figure 3. Observing how optimal hyperparameter choices may change with even larger datasets is an interesting experiment that we plan to conduct in the future, after collecting larger datasets.
>
>
> - “I would also revise L6 in the Discussion section by adding caution to the statement that the results transfer to real-world settings, especially because this is conditioned on the difficulty of a task.”
>
> We agree, and have revised the language in the updated manuscript (see red text in section 4.7) to be more clear that “design decisions made in simulation can potentially transfer to real world settings”.
>
>
> - “The paper would benefit by detailing what about a task makes it difficult in this context, is it the long-horizon factor, is it the precision factor, is it multi-arm (agent) factor, or is it all of the above, and are there other aspects of a task that could make it difficult for these learning tasks?”
>
> This is a great point. We give some intuition below for what makes the real-world tasks we evaluated difficult and how they form a spectrum for aspects of difficulty. Each of these tasks has certain bottleneck states that require precise sequences of actions -- the precision required at such states often dictates the task difficulty. Lift is the simplest task and just requires picking the small block. The bottleneck here is successfully grasping the block, and it is relatively tolerant to errors in precision. The Can task is representative of pick-and-place tasks (a very common class of task), and requires a more precise grasp than Lift due to the shape and material of the can, which makes it significantly more difficult than Lift. Tool Hang is our hardest task due to the multiple task bottlenecks (assembling the tool stand, and hanging the tool), and requirement for very high precision manipulation.

---

> > ### Author Response · Authors · 2021-08-21
> > **Response (Part 2)**
> >
> > Please note that our response has been split into two parts due to space constraints. (Part 2/2)
> >
> > - “The effect of observation spaces in the performance for IL algorithms could be more profound than just low-dimension (object state estimation) vs. high-dimension (visual raw data).”
> >
> > We concede that there are several more interesting observation space variants that could have been investigated in this study (3D sensors, tactile/haptic sensors, etc.), but we felt that comparing the use of ground-truth object observations to raw image observations was one of the most important comparisons to make, since these two observation spaces are extremely common for closed-loop policy learning in robot manipulation domains. We chose to limit the scope of the study in order to feasibly carry out the large number of evaluations necessary for completing the study (5 sim + 3 real tasks, 6 algorithms, 3 dataset sources, 2 observation spaces, and several hyperparameter choices). However, we believe that our code and dataset release will make it easy for researchers and practitioners to reproduce our results and easily compare alternate observation space choices. Our code is specifically built to offer flexibility in specifying which observations are used for training. Alternate observation space choices that could be investigated in the future could include using additional modalities (depth, force) or changing how robot or object observations are represented (frame-of-reference, as you pointed out).
> >
> >
> > - “Also, what about the transport task made it difficult to be learned using the high-dimensional observation space?”
> >
> > Unlike the other tasks, the transport task involves actuating two robot arms simultaneously - this means that by default the policy must take 4 images (2 shoulder cameras and 2 wrist cameras) as input and return actions per arm. In addition, the task involves several objects (whose poses are not given), multiple stages, and requires careful coordination between the two arms during the handover portion.
> >
> >
> > - “The paper can become stronger with comments on how these variables affect the performance in the presence of covariate shift, which BC algorithms generally succumb to”
> >
> > Covariate shift is a frequent concern when learning from offline datasets. In fact, it is responsible for many of the task failures that our trained policies encountered, especially on the Square, Transport, and Tool Hang tasks, where small mistakes during insertion or handover can have disastrous outcomes due to unseen states. This was especially true for the real world Tool Hang task (see here).
> >
> >
> > - “​​Can the paper dive a little deep into what differences in the nature of the mixed quality machine-generated datasets when compared to mixed-quality human datasets make it difficult for SOTA RL algorithms such as BCQ to perform comparably with human datasets?”
> >
> > We performed an empirical analysis and compared the performance of the learning algorithms on machine-generated and multi-human data because it was the most straightforward way to see the differences between the two types of data. Doing a deeper investigation into how machine-generated data differs from human-generated data and using these insights to improve the performance of batch RL algorithms is an exciting direction for future work. We hope that our code and dataset release will make it easy for future work to conduct such investigations.

---

### Meta-Review · Area_Chair_zCXj · 2021-08-10

**Recommendation:** Accept (Oral)
**Confidence:** 5

**Metareview:**

All of the reviewers provided scores for accepting the paper. The authors addressed most of the points raised by the reviewers in their rebuttal. I encourage the authors to address the remaining points raised in the reviews for the final version of the paper.

---

> ### Author Response · Authors · 2021-08-21
> **Response to Area Chair zCXj**
>
> Please note that our response has been split into two parts due to space constraints. (Part 1/2)
>
> We thank all of the reviewers and the meta-reviewer for their detailed and insightful comments. We have made some revisions to the paper and supplementary material (highlighted in red, and pointed out in our responses to each reviewer), and we have also replied to each reviewer’s comments. We also reply to the meta-reviewer’s comments below.
>
> - “the related work section is not fully complete and it misses a few relevant citations”
>
> We have added the relevant citations - please take a look at the revised related work section (Appendix A in the supplementary material, with the changes highlighted in red). We now summarize the important differences from the related work pointed out by reviewer YubL.
>
> Lemme et al. focuses on learning point-to-point reaching motions from demonstrations while we focus on tabletop manipulation settings where the robot must interact with one or more objects. However, unlike our study, Lemme et al. also evaluates the robustness of the learned robot motions by introducing perturbations. While similar mechanisms could be used to understand the robustness of the policies trained in this study, and such evaluations are important for deploying policies in real-world settings, we leave this for future work.
>
> Similar to our study, Rana et al. collected demonstration trajectories across many humans and tasks. However, their datasets consist of robot end effector trajectories, while our datasets and learning methods focus on leveraging additional modalities such as object poses and camera images to train policies that can solve tasks across several scene configurations. Interestingly, Rana et al. also used crowdsourced humans to evaluate learned robot motions with subjective metrics such as safety, while we primarily evaluate our policies using task success rate. Subjective measures like safety are important for real-world policy deployment, and it will be interesting to study this further in the future.
>
>
> - “the study presented in the paper is limited to a single robot, which limits the potential generality of the claims”
>
> This is a great point! We chose to limit the scope of the study in order to feasibly carry out the large number of evaluations necessary for completing the study (5 sim + 3 real tasks, 6 algorithms, 3 dataset sources, 2 observation spaces, and several hyperparameter choices). However, we do indeed have some evidence that the results transfer to other robotic platforms. The IRIS paper (https://arxiv.org/abs/1911.05321) presented results on the crowdsourced RoboTurk cans dataset (https://roboturk.stanford.edu/dataset_sim.html), which consists of a Sawyer robot arm, and 100 hz joint velocity actions. The authors showed that they could achieve a success rate of 30%, which was the best reported performance from offline training on this dataset (for training on low-dimensional observations). We used our study insights to train a BC-RNN model that achieves over 60% success rate (double the performance!). Note that this dataset contains a different robot (Sawyer instead of Franka) and a different action space (100 hz joint velocity instead of 20 hz operation space end effector control).
>
> Also, our results on the two arm transport task suggest that our insights may transfer to bimanual robot morphologies, where two arms must be controlled simultaneously.
>
> Furthermore, we do not see any theoretical reasons why the general observations we made about learning from human datasets would not extend to similar types of robot arms (in our case, 7 degree-of-freedom arms with a stationary camera observing the scene and another camera attached to the end-effector). Differences in morphology may of course affect the results, but we believe that they would not affect the general conclusions we drew. This would be best supported by further experimental evaluation.
>
> Finally, the robosuite simulation framework, in which we developed our simulation tasks, easily supports replacing the Franka arm with other arms such as the Sawyer, and further supports the collection of human demonstrations via keyboard and 3d mouse interfaces. Our released code will be compatible with human demonstrations collected through the robosuite framework, making it easy for other researchers to explore datasets that involve other robotic arms.
>
>
> - “it would be desirable to discuss additional aspects of the datasets such as the frame-of-reference of the sensory readings as well as their dimensionality”
>
> While we have omitted the specific details of the sensory readings in the main text, Appendix E (in the supplementary material) contains all of these details for both the low-dimensional and image observation spaces, for each task. We also added a note in the revised main text (see section 3.2, “Observation Modalities” paragraph) to refer readers to this section of the appendix for full details.

---

> > ### Author Response · Authors · 2021-08-21
> > **Response (Part 2)**
> >
> > Please note that our response has been split into two parts due to space constraints. (Part 2/2)
> >
> > - “additional insights on the differences between human-generated and machine-generated data would be informative for the community”
> >
> > We performed an empirical analysis and compared the performance of the learning algorithms on machine-generated and multi-human data because it was the most straightforward way to see the differences between the two types of data. Doing a deeper investigation into how machine-generated data differs from human-generated data and using these insights to improve the performance of batch RL algorithms is an exciting direction for future work. We hope that our code and dataset release will make it easy for future work to conduct such investigations.

---

> > > ### Comment · Area_Chair_zCXj · 2021-08-24
> > > **Thank you for the response**
> > >
> > > Thank you for addressing the points raised by me and other reviewers - this is very helpful in determining the final decision for the submission. I particularly appreciate the thorough responses that further clarify the contributions of the paper.

---

### Decision · Program_Chairs · 2021-09-13

**Decision:**

Accept (Oral)

**Comment:**

All of the reviewers provided scores for accepting the paper. The authors addressed most of the points raised by the reviewers in their rebuttal. I encourage the authors to address the remaining points raised in the reviews for the final version of the paper.